

# Identification and characterization of auxin response factor (ARF) family members involved in fig (*Ficus carica* L.) fruit development

Haomiao Wang[1], Hantang Huang[1], Yongkai Shang[1], Miaoyu Song[1] and Huiqin Ma[1,2]

[1] College of Horticulture, China Agricultural University, Beijing, Beijing, China
[2] State Key Laboratory of Agrobiotechnology, China Agricultural University, Beijing, Beijing, China

Corresponding authors
Miaoyu Song,
songmiaoyu@cau.edu.cn
Huiqin Ma, hqma@cau.edu.cn

## ABSTRACT

The auxin response factor (ARF) combines with AuxREs cis-acting elements in response to auxin to regulate plant development. To date, no comprehensive analysis of ARF genes expressed during fruit development has been conducted for common fig (*Ficus carica* L.). In this study, members of the FcARF gene family were screened, identified in the fig genome database and their features characterized using bioinformatics. Twenty FcARF genes were clustered into three classes, with almost similar highly conserved DBD (B3-like DNA binding domain), AUX/IAA (auxin/indole-3-acetic acid gene family) and MR domain structure among class members. Analysis of amino acid species in MR domain revealed 10 potential transcription activators and 10 transcription inhibitors, and 17 FcARF members were predicted to be located in the nucleus. DNA sequence analysis showed that the ARF gene family consisted of 4–25 exons, and the promoter region contained 16 cis-acting elements involved in stress response, hormone response and flavonoid biosynthesis. ARF genes were expressed in most tissues of fig, especially flower and peel. Transcriptomics analysis results showed that *FcARF2*, *FcARF11* and *FcARF12*, belonging to class-Ia, were stably and highly expressed in the early development stage of flower and peel of 'Purple peel' fig. However, their expression levels decreased after maturity. Expression of class-Ic member *FcARF3* conformed to the regularity of fig fruit development. These four potential transcription inhibitors may regulate fruit growth and development of 'Purple Peel' fig. This study provides comprehensive information on the fig ARF gene family, including gene structure, chromosome position, phylogenetic relationship and expression pattern. Our work provides a foundation for further research on auxin-mediated fig fruit development.

## INTRODUCTION

Auxin is an important plant hormone, which directly regulates cell growth, division and specific differentiation, participating in embryogenesis, flower and fruit development, root

formation and vascular bundle development (*Du et al., 2012*). At the molecular level, auxin regulates plant growth and development *via* a signal transduction pathway mediated by Aux/IAA proteins and auxin response factor (ARF) (*Liu et al., 2015*; *Woodward et al., 2005*). ARFs typically consist of three domains: an N-terminal B3-like DNA binding domain (DBD), a C-terminal dimerization domain (CTD), which forms a heterodimer with Aux/IAA family genes, and the middle domain (MR) with an activation (AD) or inhibition (RD) role (*Guilfoyle & Hagen, 2007*; *Kim, Harter & Theologis, 1997*; *Li et al., 2015*). The DBD domain can identify Auxin Response Element (AuxRE, 'TGTCTC') in auxin response gene promoter. The CTD of ARF protein contains two conserved motifs, motif III and motif IV, which facilitate formation of homo- and hetero-dimerization among ARFs and Aux/IAAs (*Zhou et al., 2017*). The MR domain either activates or inhibits transcription depending on its amino acid composition (*Ye et al., 2020*). Therefore, these domains are necessary for efficient regulatory function of ARF (*Wang et al., 2007*).

The ARF gene family has been identified in many plants. There are 23 ARFs in *Arabidopsis thaliana* (*Okushima et al., 2005*), 12 in *Fragaria vesca* (*Wang et al., 2019*), 19 in *Vitis vinifera* (*Wan et al., 2014*), 19 in *Citrus sinensis* (*Li et al., 2015*), 19 in *Capsicum annuum* L. (*Zhang et al., 2017*), 22 in *Lycopersicon esculentum* (*Kumar, Tyagi & Sharma, 2011*), 25 in *Oryza sativa* L. (*Wang et al., 2007*) and 37 in *Populus alba × P. glandulosa* (*Yang et al., 2014*). Based on evolutionary relationship analysis, ARF genes in *Arabidopsis thaliana* cluster into three classes. Class I is the largest with 15 members belonging to three classes. Members in class-Ia are mainly involved in regulating flower and root development. For example, *AtARF1* and *AtARF2* promote flower senescence and abscission (*Ellis et al., 2005*), whereas *AtARF11* promotes lateral root formation (*Feng et al., 2012*). These genes share a high degree of similarity among their amino acid and nucleotide sequences. In addition, *AtARF23* is a pseudogene (*Okushima et al., 2005*). Class-Ic has two members, *AtARF3* and *AtARF4*, which regulate floral organogenesis and the timing and pattern of development (*Ellis et al., 2005*; *Harper et al., 2000*; *Hunter et al., 2006*; *Pekker, Alvarez & Eshed, 2005*; *Sessions et al., 1997*). Members of class-II are ARF^ClassA transcription activators, which regulate auxin-mediated flower development, morphogenesis, lateral root formation and other processes (*Truskina et al., 2021*). Class-III consists of three most structurally distinct members that regulate pollen wall pattern formation, root cap formation or seed germination (*Ellis et al., 2005*; *Guilfoyle & Hagen, 2007*).

As important plant growth regulators, ARF genes are expressed and play regulatory roles in different tissues (Tables S1 and S2), especially floral organs and fruit. *AtARF2* was found to promote the development of stamens and participate in the senescence and abscission of floral organs (*Okushima et al., 2005*). *AtARF4* regulates flower patterns and plays a significant role in organ polarity development (*Hunter et al., 2006*; *Liu et al., 2014*; *Pekker, Alvarez & Eshed, 2005*). *AtARF6/8* and *SlARF6/8* can promote maturation of floral organs and regulate fruit and seed carpel development (*Goetz et al., 2006*; *Wu, Tian & Reed, 2006*). In fruiting plants, *SlARF3* plays multiple roles in tomato fruit development and participates in the formation of epidermal cells and trichomes (*Zhang et al., 2015*). *SlARF9* regulates cell differentiation during early tomato fruit development and *SlARF4*

controls sugar metabolism during maturation (*de Jong et al., 2015*; *Sagar et al., 2013*). Six CiARF genes (*CiARF1/2/6/7/12/18*) are highly expressed in fruit receptacle, where they regulate the citrus pulp and peel development (*Li et al., 2016*). Seven PpARF genes (*PpARF1/2/3/4/5/6/9*) perform similar functions in *Prunus persica* (*Hou et al., 2021*). Studies on apples show that *MdARF5* induces ethylene synthesis by directly promoting the expression of ethylene synthesis genes *MdACS3a*, *MdACS1* and *MdACO1*, which then promote apple fruit ripening (*Yue et al., 2020*).

ARF proteins are negatively regulated by auxin/indole-3-acetic acid (Aux/IAA) proteins. The potential for interaction between ARF and Aux/IAA proteins is high as their C-terminal domains are homologous (*Shen et al., 2010*). Aux/IAA genes play a major role in regulating many auxin processes in plants. Aux/IAA genes exist as a family in many plants and are associated with leaf curl patterns, plant phototropism and tropism, plant height, and especially lateral root formation (*Reed, 2001*). In *Arabidopsis thaliana*, *axr5/iaa1* mutant showed auxin resistance and loss of root and stem geotropism and hypocotyl phototropism (*Yang et al., 2004*). Mutations in *AUXIN RESISTANT1* (*AUX1*)/ *LIKE-AUX1* (*LAX*) genes result in auxin-related developmental defects and have been implicated in regulating root development, female gametophyte development and embryo development in *Arabidopsis thaliana* (*Swarup & Bhosale, 2019*). *AtIAA8* regulates lateral root formation by interacting with auxin receptor *AtTIR1* and ARF transcription factors in the nucleus (*Arase et al., 2012*). *MdARF1* regulates anthocyanin accumulation in fruit by interacting with *MdIAA121* in apples (*Wang et al., 2018*).

As an important plant growth regulator, the role of ARF gene family in fruit ripening and development has been investigated in many species but not in fig fruits. With its origins in the Mediterranean coastal region, the common fig (*Ficus carica* L.) is one of the widely cultivated ancient fruit trees in the world. The fig fruit (syconia), an aggregated fruit formed by the expansion of female flower tissue and receptacle, has high antioxidant activity, great taste and high nutritional value (*Song et al., 2021*). The fig fruit exhibits a typical double sigmoid growth curve, including two rapid growth phase (I and III) separated by slower growth phase (II) (*Song et al., 2021*). The fig has the advantages of short juvenile period, easy vegetative growth and reproduction, and is gradually becoming a new model plant for studying fruit development mechanisms (*Wang et al., 2018*). Presently, the genome-wide characterization of fig auxin signal transduction-related components and their expression patterns has not been carried out. In this study, genome-wide data of 'Horaishi' and 'Dottato' (*Mori et al., 2017*; *Usai et al., 2020*) and 'Purple Peel' fig transcriptome data at different developmental stages (NCBI accession number PRJNA723733) were used to predict ARF genes. In total, 20 ARF genes were identified in common fig, and their features, including chromosome position, phylogenetic relationship, gene structure and motif characteristics, characterized. In addition, we described the transcriptional accumulation patterns of ARF genes in different tissues, including at different developmental and maturation stages of female flowers and peel. Our study provides a good basis for further exploring the role of *FcARFs* and characterizing their functions.

## MATERIALS AND METHODS

### Plant materials

The fig cultivar 'Purple Peel' plants were established at Shangzhuang Experimental Station of China Agricultural University, Beijing, China. The roots, stems, leaves of 'Purple Peel', and fruits at different developmental stages were collected for qRT-PCR analysis. Fruits were sampled at six stages based on the characteristics of fruit development. Stage 1 represented phase I (the first rapid growth period), stages 2, 3, and 4 were the early, middle, and late stages of phase II (slow growth period), and stages 5 and 6 represented phase III (the second rapid growth period). At each stage, 50 fruits were sampled. The peel and flower were separated onsite during sampling. Fresh samples were quick-frozen with liquid nitrogen and stored at −80 °C until RNA extraction.

### Identification of ARF transcription factor family in fig

Genome sequences of *cv.* Horaishi and *cv.* Dottato fig were downloaded from the National Center for Biotechnology Information (NCBI) (https://www.ncbi.nlm.nih.gov/genome/?term=Ficus+carica) (*Mori et al., 2017*; *Usai et al., 2020*). The sequences were blasted (E-value = −5) using the hidden Markov model (HMM) files of B3, Auxin_resp and AUX_IAA domains in Pfam (http://pfam.xfam.org/search/sequence) with entry numbers PF02362, PF06507 and PF02309, respectively (*El-Gebali et al., 2019*). Candidate genes containing known conserved domains were retained, and further screened for ARF domains against three databases: Pfam, NCBI conserved domains (http://www.ncbi.nlm.nih.gov/Structure/cdd/wrpsb.cgi) and SMART (http://smart.emblheidelberg.de) (*Letunic, Khedkar & Bork, 2021*). ARF family genes in *Arabidopsis thaliana* were downloaded from the Arabidopsis database (TAIR; https://www.Arabidopsis.org/). Fig ARF homologs were compared with Arabidopsis ARFs using BLASTP with default parameters to obtain annotation and grouping information. *FcARFs* sequences were analyzed using bioinformatics methods, and the physicochemical parameters, MW, pI and GRAVY of translated proteins were calculated using ExPASy (http://www.expasy.ch/tools/pi_tool.html) (*Guo et al., 2014*).

### Multiple sequence alignment and phylogenetic analysis

Using Clustal 2.0, multiple-sequence alignments of amino acid sequences of predicted ARF genes in fig was carried out, with default parameters (*Song et al., 2021*). ClustalX 2.11 was used to analyze phylogenetic relationships among ARF gene families in Arabidopsis, fig, apple, grape and strawberry. (*Liu et al., 2015*; *Luo et al., 2014*; *Sun, Fan & Ling, 2015*). The phylogenetic tree of *FcARFs* was constructed using MEGA 7.0 *via* the Neighbor-Joining method with calibration test parameter "Bootstrap = 1,000 (1,000 replicates)" (*Tamura et al., 2011*; *Kumar, Stecher & Tamura, 2016*). The analysis involved 87 amino acid sequences and ARF gene families in five species including fig were divided following the division of ARF gene families in *Arabidopsis thaliana*. Percentage confidence values were shown on branches and the evolutionary distances used to infer the phylogenetic tree *via* Poisson correction method (*Zuckerkandl & Pauling, 1965*). Gaps were treated as a complete deletion. There were a total of 1,619 positions in the final dataset.

## Gene structure and conserved motif analysis

*FcARFs* gene sequences were downloaded from fig genome database (https://www.ncbi.nlm.nih.gov/genome/?term=Ficus+carica), and the intron/exon structure map of the fig ARFs was generated online using the Gene Structure Display Server (GSDS: http://gsds.cbi.pku.edu.ch) (*Guo et al., 2007*). FcARFs amino acid sequences were referenced to fig genome database protein RefSeq. Because ARF family genes contain highly conserved domains, DNAMAN was used for multiple sequence alignment of FcARF protein sequences. The length and sequence information of other conserved motifs except the ARF conserved domain were obtained using the online MEME4.11. 2 tool (http://meme.sdsc.edu/meme/itro.html) with the following parameters: number of repetitions "any," highest motif number "15," motif length "6–100," and default values for the other parameters.

## Chromosome mapping and collinearity analysis

Tandem replication of ARF genes usually refers to adjacent homologous genes on the same chromosome, whereas gene duplication events are regarded as repeated homologous genes on different chromosomes (*Liu et al., 2011*). Whether *FcARF* gene is considered an important event of tandem repeat between genes depends on its location on the chromosome. To study the distribution of ARF genes on chromosomes, their chromosomal location and annotation information were utilized. An openly available database and Map Draw software were used to generate physical location information for homolinear analysis of *ARF* genes in fig. The genome data of *Ficus hispida* and *Ficus microcarpa* were downloaded from the National Genomics Data Center database (https://ngdc.cncb.ac.cn/search/?dbId=bioproject&q=PRJCA002187&page=1) (*Zhang et al., 2020*). An interspecies collinearity analysis of ARFs in fig, *F. hispida*, and *F. macrocarpa* was performed using MCscanX and TBtools (*Song et al., 2021*). The final map was generated with Circos version 0.63 (http://circos.ca/). The non-synonymous replacement rate (Ka) and synonymous replacement rate (Ks) of replicated gene pairs were calculated using KaKs_Calculator 2.0 (*Wang et al., 2010*), with Ka/Ks ratio taken as environmental selection pressure.

## Protein functional connection network

Since fig species was not included in STRING database, the search species was set to *Arabidopsis thaliana*. The interaction networks of 20 ARF proteins were analyzed using the STRING protein interaction database (http://string-db.org/), which is an online protein-protein interaction networks functional enrichment analysis prediction site (*Szklarczyk et al., 2021*). Protein-protein interaction networks were built with an interaction score of −1 to 1. In total, 20 fig ARF proteins were aligned with AtARF proteins using Blastp alignment with E-value of 1e−0 (*Rosas-Camacho, 2009*; *Chen et al., 2015*). The interaction network of FcARF protein was constructed using homologous and *Arabidopsis thaliana* ARF protein, and genes of interest were visualized using Cytoscape software (*Doncheva et al., 2019*).

### Expression analysis of *FcARFs* using RNA-seq and qRT-PCR

TBtools was used to analyze the expression patterns of FcARFs in 'Purple-Peel' fig fruit based on RNA-seq library (NCBI accession number PRJNA723733). Their peel and female flower tissue were collected at six stages of fruit development and significant differential expression was determined with $p < 0.05$ and |log2(fold change)| ≥ 1.

Different tissues and fruits were sampled from 'Purple Peel' plant at six stages for qRT-PCR. The peel and flower were separated during sampling. Fresh samples were quick-frozen with liquid nitrogen and stored at −80 °C for RNA extraction. Total RNA was extracted using the CTAB method and reversed transcribed using reverse transcription kit (TaKaRa, Dalian, China). The specific primers for qRT-PCR are shown in Table S5. The design parameters for the primers include amplicon length, 100–150 bp; primer length, 15–25 bp; melting temperature (Tm), 57–61 °C; GC base content, 40–60%; and excluding conserved domain (*Li et al., 2019*). Following a previously published method (*Song et al., 2021*), all qRT-PCR reaction volumes were set at 10 μL. The cycle parameters were as follows: pre-denaturation at 94 °C for 15 s, annealing at 60 °C for 30 s and extension at 72 °C for 1 min, for 40 cycles. qRT-PCR reaction for each gene was replicated three times. *Fc18S* and *FcActin* were used as an internal standard to calculate relative fold differences based on comparative cycle threshold values. We calculated relative expression levels using the comparative Ct ($2^{-\Delta\Delta Ct}$) method. SPSS 26.0 software was used for significance analysis.

## RESULTS

### ARF family members identified in fig

Twenty ARF genes were identified in published fig genome and named *FcARF1* to *FcARF20* following the equivalent classification for *Arabidopsis thaliana* (Fig. 1A and Table 1). Detailed information for each candidate gene, including gene name, gene ID, number of exons, protein length, protein molecular weight (kDa), isoelectric point (pI), instability coefficient, Aliphatic index, GRAVY and location in cells, is shown in Table 1. The length of amino acids encoded by *FcARFs* ranged from 588 (*FcARF10*) to 1,150 (*FcARF7*), averaging 782.95 amino acids. The number of exons and predicted molecular weight ranged from three (*FcARF10*) to 17 (*FcARF18*) and 52.73 (*FcARF4*) to 127.06 KDa (*FcARF7*), respectively. The isoelectric point (pI) ranged from 5.03 (*FcARF5*) to 9.52 (*FcARF11*), and acidic amino acids were enriched in the protein molecules. In total, 80.95% of fig ARF family proteins had isoelectric points less than seven, with most of them being weakly acidic, consistent with pI of ARF proteins in citrus and strawberry (*Li et al., 2015*; *Wang et al., 2019*). On average, the total hydrophilicity of ARF family proteins in fig was negative, ranging from −0.726 (*FcARF11*) to −0.322 (*FcARF17*), which indicated that these proteins were hydrophilic. The instability index ranged from 42.05 (*FcARF17*) to 72.35 (*FcARF15*), suggesting that all these proteins were stable (II > 40). The Aliphatic index ranged from 63.06 (*FcARF11*) to 84.46 (*FcARF18*). Twenty FcARF proteins were localized in subcellular structures by online software Target P and CELLO v2.5. The results showed that they were mostly distributed in the nucleus and cytoplasm.

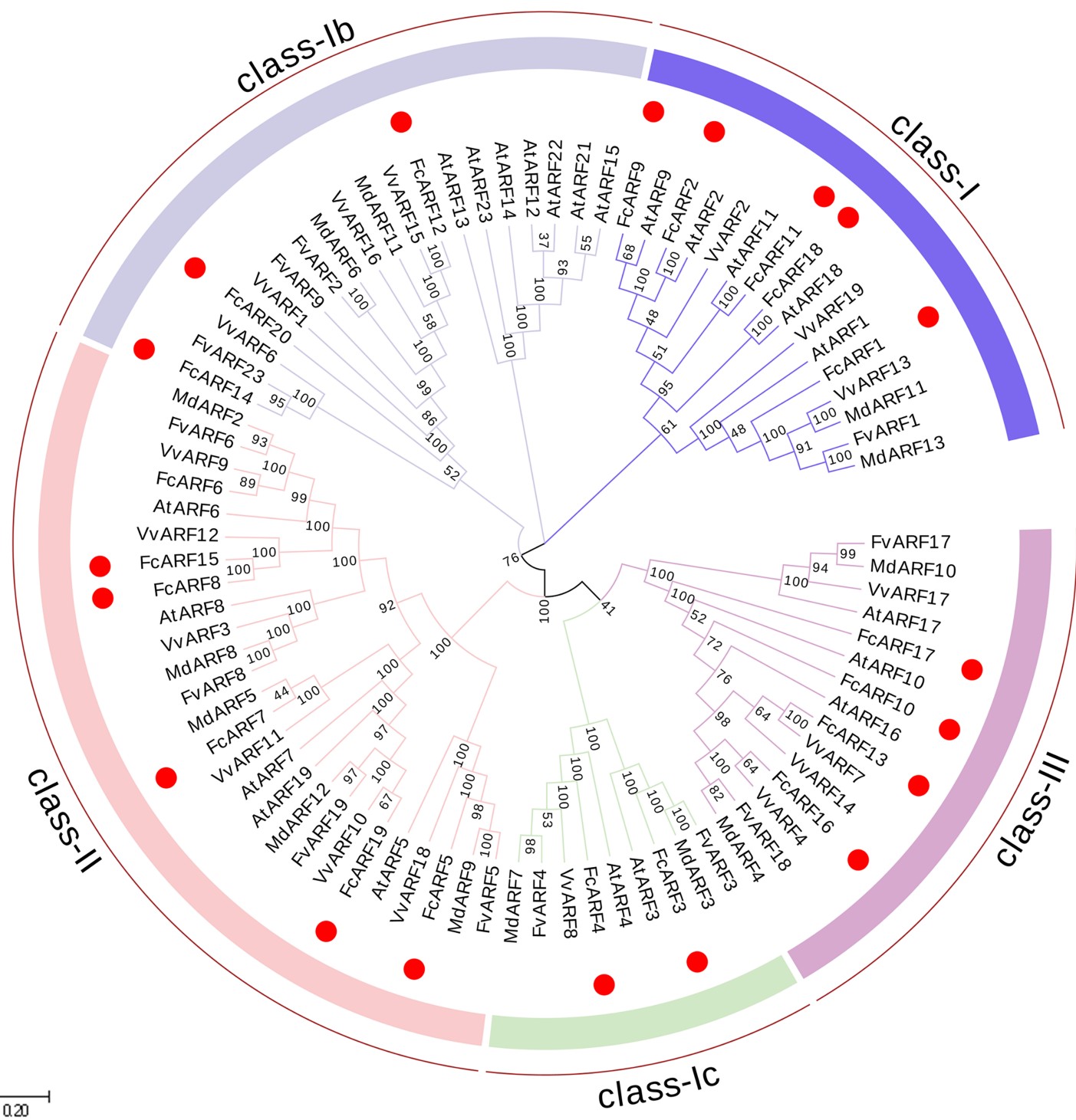

**Figure 1 Phylogenetic analysis of auxin response factors from *Ficus carica* L., *Arabidopsis thaliana*, *Fragaria vesca*, *Vitis vinifera* and *Malus domestica*.** The phylogenetic tree was constructed using the neighbor-joining method in MEGA 7.0. Bootstrap values from 1,000 replicates were indicated at each branch. Gene groups were indicated with different colors and were classified into four groups: class-I, class-II and class-III. class-I includes three subclasses: class-Ia class-Ib and class-Ic.

**Table 1 FcARF genes encoding ARF proteins along with their molecular details.**

| Gene name | Gene ID | Exon No. | Length (AA)[a] | Mw (kDa)[b] | Atomic formula | $pI$[c] | Instabiliy index | Aliphatic index | GRAVY | Localization |
|---|---|---|---|---|---|---|---|---|---|---|
| *FcARF1* | FCD_00029581 | 16 | 897 | 99.27 | $C_{3279}H_{5126}N_{928}O_{1019}S_{34}$ | 6.04 | 71.85 | 73.03 | −0.463 | Nuclear |
| *FcARF2* | FCD_00016093 | 14 | 843 | 94.02 | $C_{4116}H_{6423}N_{1191}O_{1271}S_{34}$ | 6.72 | 55.25 | 64.06 | −0.654 | Nuclear |
| *FcARF3* | FCD_00001359 | 12 | 813 | 89.96 | $C_{3946}H_{6167}N_{1111}O_{1229}S_{35}$ | 6.55 | 56.15 | 71.5 | −0.465 | Nuclear |
| *FcARF4* | FCD_00032414 | 11 | 768 | 52.73 | $C_{2335}H_{3623}N_{647}O_{714}S_{17}$ | 7.21 | 57.64 | 69.49 | −0.510 | Nuclear |
| *FcARF5* | FCD_00003481 | 15 | 942 | 103.77 | $C_{4544}H_{7139}N_{1241}O_{1448}S_{45}$ | 5.03 | 58.47 | 78.33 | −0.370 | Nuclear |
| *FcARF6* | FCD_00013415 | 14 | 987 | 99.27 | $C_{4360}H_{6798}N_{1234}O_{1347}S_{38}$ | 6.04 | 71.85 | 73.03 | −0.463 | Nuclear |
| *FcARF7* | FCD_00017652 | 13 | 1,150 | 127.056 | $C_{5531}H_{8723}N_{1595}O_{1737}S_{53}$ | 6.57 | 63.56 | 73.5 | −0.511 | Nuclear |
| *FcARF8* | FCD_00034325 | 14 | 899 | 99.39 | $C_{4367}H_{6781}N_{1229}O_{1362}S_{34}$ | 5.86 | 71.46 | 72.45 | −0.491 | Nuclear |
| *FcARF9* | FCD_00028996 | 14 | 687 | 76.74 | $C_{3366}H_{5311}N_{949}O_{1054}S_{25}$ | 6.22 | 52.53 | 72.63 | −0.523 | Nuclear |
| *FcARF10* | FCD_00021037 | 3 | 588 | 65.32 | $C_{2901}H_{4478}N_{806}O_{871}S_{23}$ | 6.95 | 43.7 | 70.1 | −0.475 | Nuclear |
| *FcARF11* | FCD_00022972 | 11 | 595 | 67.15 | $C_{2969}H_{4638}N_{870}O_{878}S_{1}$ | 9.52 | 60.21 | 63.06 | −0.726 | Nuclear |
| *FcARF12* | FCD_00032300 | 12 | 689 | 78.08 | $C_{3453}H_{5432}N_{1012}O_{1020}S_{20}$ | 9.42 | 54.24 | 72.12 | −0.655 | Nuclearn |
| *FcARF13* | FCD_00005247 | 5 | 699 | 76.48 | $C_{3378}H_{5233}N_{945}O_{1025}S_{31}$ | 6.36 | 50.71 | 71.43 | −0.390 | Cytoplasmic |
| *FcARF14* | FCD_00007016 | 14 | 721 | 81.01 | $C_{3572}H_{5504}N_{1000}O_{1081}S_{39}$ | 6.35 | 60.08 | 66.81 | −0.519 | Nuclear |
| *FcARF15* | FCD_00016691 | 14 | 902 | 99.73 | $C_{4381}H_{6807}N_{1233}O_{1368}S_{34}$ | 5.83 | 72.35 | 72.64 | −0.493 | Nuclear |
| *FcARF16* | FCD_00001287 | 4 | 708 | 77.28 | $C_{3399}H_{5310}N_{972}O_{1031}S_{31}$ | 7.62 | 44.78 | 72.57 | −0.366 | Cytoplasmic |
| *FcARF17* | FCD_00001289 | 4 | 620 | 67.06 | $C_{2960}H_{4639}N_{819}O_{930}S_{15}$ | 5.90 | 42.05 | 79.69 | −0.322 | Nuclear |
| *FcARF18* | FCD_00035227 | 17 | 617 | 69.43 | $C_{3054}H_{4812}N_{862}O_{921}S_{34}$ | 6.04 | 56.32 | 84.46 | −0.326 | Cytoplasmic |
| *FcARF19* | FCD_00010593 | 13 | 1,115 | 124.55 | $C_{5423}H_{8466}N_{1576}O_{1712}S_{44}$ | 6.03 | 70.65 | 68.66 | −0.704 | Nuclear |
| *FcARF20* | FCD_00013930 | 14 | 719 | 80.78 | $C_{3576}H_{5568}N_{1014}O_{1074}S_{26}$ | 6.28 | 51.39 | 75.26 | −0.554 | Nuclear |

Notes:
[a] Length of open reading frame in base pairs.
[b] Molecular weight (Kilodaltons).
[c] Isoelectric point.

## Phylogenetic analysis of FcARFs

In total, 87 ARF proteins comprising 23 from *Arabidopsis thaliana*, 20 from fig, 12 from strawberry, 19 from grape and 13 from apple were obtained from NCBI database to construct a phylogenetic tree using Neighbor-Joining approach. FcARF proteins were divided into three classes (Fig. 1). Class-I was further divided into three classes, including class-Ia with the following five members: *FcARF1*, *FcARF2*, *FcARF9*, *FcARF11* and *FcARF18*. Class-Ib included three members (*FcARF12*, *FcARF14* and *FcARF20*) whereas class-Ic included two (*FcARF3* and *FcARF4*). Class-II had six members and class-III had four. Compared with the equivalent *Arabidopsis* class, the number of ARF members in fig class-Ib was less. These results indicated that gene loss or gain occurred during the evolution of ARF family genes in fig.

## Analysis of ARF gene family structure

Typical ARF protein has three conserved domains: N-terminal B3-like DNA binding domain (DBD), C-terminal dimerization domain (CTD), and a middle region (MR) (*Xing et al., 2011*). Figure 2 shows that all FcARFs contained MR as an intermediate domain.

All proteins contained a complete CTD structure except class-III members. Similarly, DBD structure existed in all FcARFs proteins except *FcARF18*. The branches of the same phylogenetic tree always had similar structures, but some genes in the same class exhibited different gene structures. Among them, four FcARFs (*FcARF17*, *FcARF10*, *FcARF13* and *FcARF16*) had truncated CTD domains distributed in class-III of the phylogenetic tree (Fig. 2A). To exhaustively verify the accuracy of FcARF conserved domains, the amino acid sequence alignment results of FcARFs and identified the location of conserved domain were aligned using DNAMAN software (Fig. S1).

The active MR domain has been found to be rich in glutamine (Q) whereas the inhibitory domain is rich in proline (P), serine (S) and threonine (T) (*Wu et al., 2011*; *Tiwari, Hagen & Guilfoyle, 2003*). Considering these characteristics, we analyzed the amino acid sequence and composition of the MR domain (Fig. S2). Subsequently, we predicted 10 potential transcription activators (*FcARF4*, *FcARF5*, *FcARF7*, *FcARF9*, *FcARF10*, *FcARF13*, *FcARF16*, *FcARF17*, *FcARF18* and *FcARF19*) and 10 inhibitors (*FcARF1*, *FcARF2*, *FcARF3*, *FcARF6*, *FcARF8*, *FcARF11*, *FcARF12*, *FcARF14*, *FcARF15* and *FcARF20*) (Fig. 2B).

## Conservative FcARFs motif and cis-acting element analysis

The exon-intron structures of genes in the same gene family are similar, reflecting the close evolutionary relationship among all its members. The number of introns in *FcARF* ranged from three to 17 (Fig. 3A). *FcARF18* gene in class-Ia had the most introns (17), whereas class-III members contained the least, suggesting that members have experienced intron loss during evolution. Conserved motifs 1–15 of *FcARFs* were analyzed (Table S3, Fig. S3). Motif 2 and motif 8 belonging to DBD and MR domains, respectively, were highly conserved in all FcARF proteins. Our analysis of conserved motifs in the database revealed that conserved motifs in FcARF are unstable, and their positions had changed except motif 2 and motif 8. In addition, motifs 15 only existed in specific genes, which agrees with previously reported results (*Wei et al., 2016*). This indicates that these motifs may be essential for specific regulatory functions of these genes.

We found 2 kb sequence upstream of the initiation codon of *FcARF* gene in PLACE database that predicted many cis-acting elements larger than 4 bp. TGA-element and TGA-box were associated with auxin response and transcriptional activation. There were at least 16 cis-acting regulatory elements in the promoter region of *FcARF* family members in fig (Fig. 3B). They were involved in abiotic stress responses, including light response, anaerobic inducibility, anoxic specific inducibility, defense and stress response, drought induction, low temperature responsive and wound response. In addition, these cis-acting regulatory elements were involved in hormone response, (abscisic acid response, auxin response, gibberellin response, MeJA response and salicylic acid response), circadian control, flavonoid biosynthesis regulation, meristem expression and other regulatory elements. Drought stress-related MYBHv1 binding sites were found in the promoter regions of three ARF genes (*FcbARF2*, *FcARF14* and *FcARF18*).

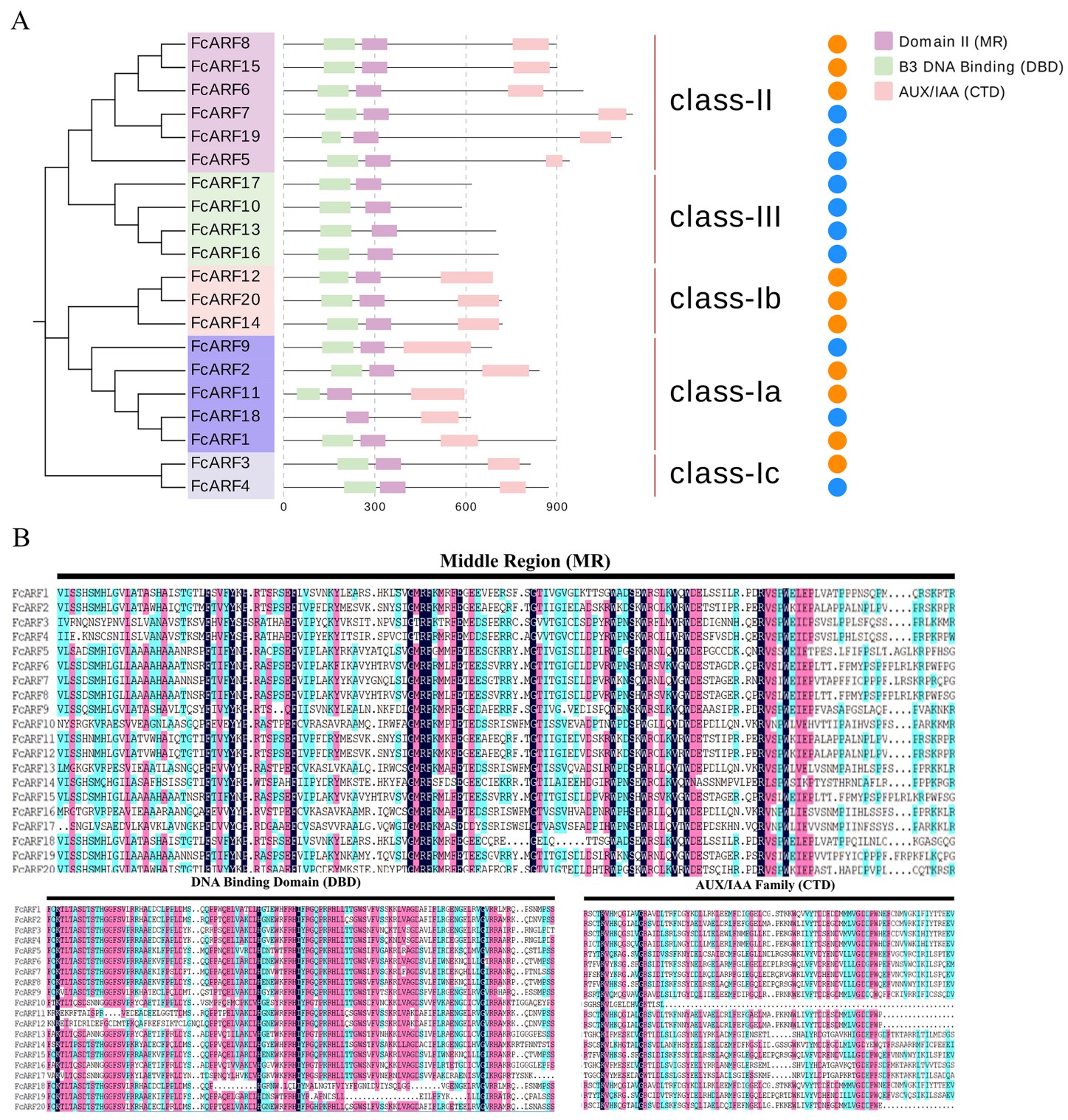

**Figure 2 Analysis of the sequence and structure of the ARF gene family in Fig.** (A) Analysis of protein domains and gene structures. The B3 DNA binding domain (DBD), middle region (MR), and AUX/IAA (CTD) family domain were shown in green, purple and pink, respectively. The orange circle represents a transcription activator, and the blue circle represents a transcription inhibitor. (B) Distribution of the conserved motifs of FcARFs. Sequence alignment of the conserved B3 DNA binding domain (DBD), middle region (MR), and AUX/IAA (CTD) were performed by CLUSTALW.

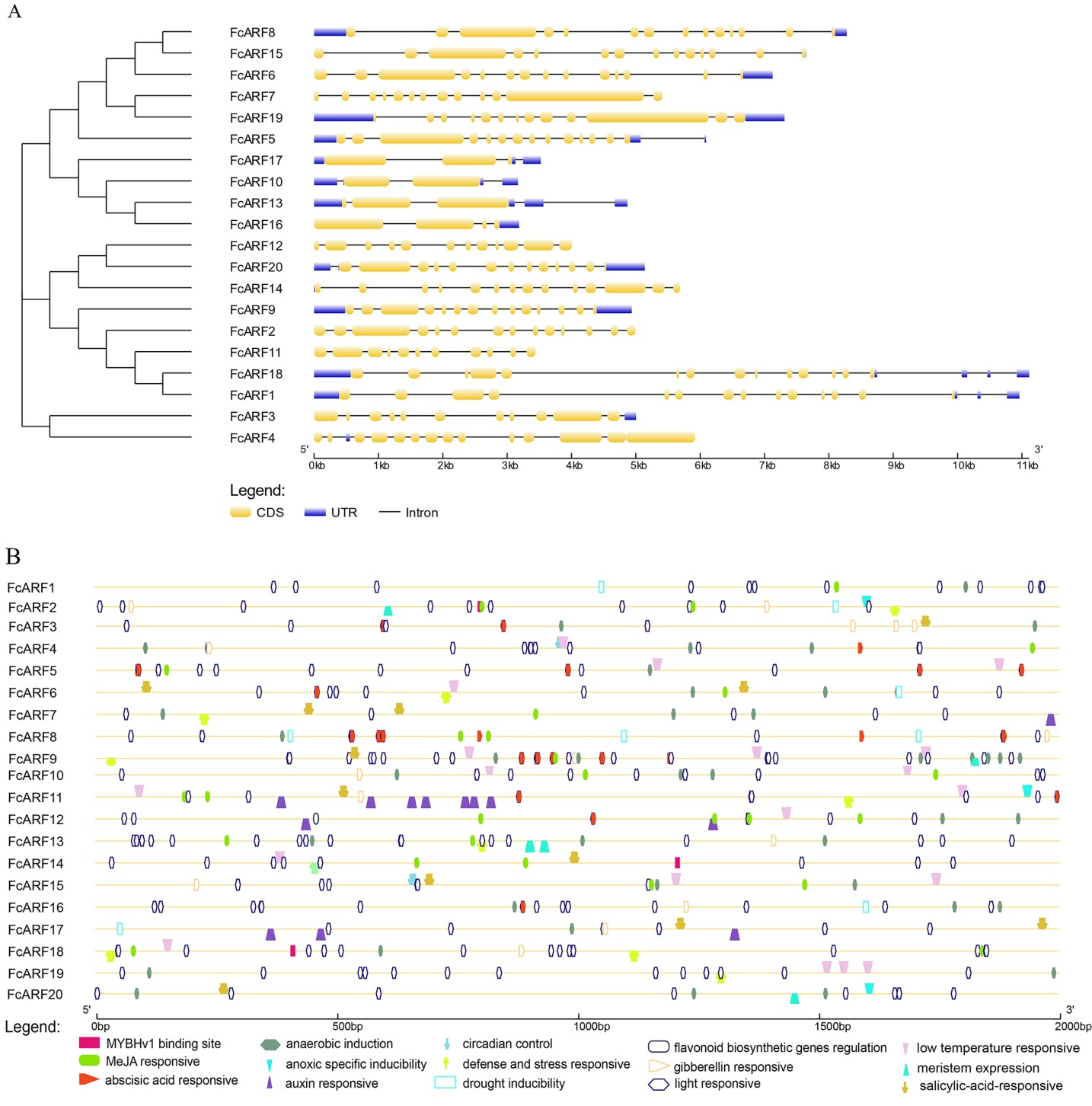

**Figure 3  Analysis of gene structures and cis-acting elements for FcARF genes.** (A) Analysis of exon-intron structures for FcARF genes. Blue rectangles, yellow rectangles and blank lines were indicated upstream/downstream, exons and introns, respectively. The lengths of exons and introns are indicated by the scale bar. (B) The distribution of cis-elements in the 2,000 bp upstream promoter regions of FcARF genes were analyzed. The *cis*-elements were predicted at PlantCARE (https://bioinformatics.psb.ugent.be/webtools/plantcare/html/). Different*cis*-elements were presented with different shapes and colors.

## Chromosome localization and distribution of FcARFs

Distributions of ARF transcription factors in fig linkage groups are shown in Fig. S4. *FcARF1–FcARF20* were unevenly distributed in 12 chromosomes, apart from chromosomes 2 and 4. Chromosomes 1, 6, 7, 10, 11 and 12 had one ARF gene each, which were *FcARF6*, *FcARF7*, *FcARF5*, *FcARF9*, *FcARF14* and *FcARF20*, respectively. Chromosomes 3, 5 and 9 had 3 ARF genes each, which were the most abundant. Tandem replication occurs when the distance between genes is less than 100 kb and exists on the same chromosomes, only FcARF3 and FcARF16 on Chr3 occurred. The FcARF gene family had three pairs of genes from fragment replication (Fig. 4A). During intraspecific evolution, the Ka/Ks values can explain positive selection (Ka/Ks > 1), neutral selection (Ka/Ks = 1) and negative selection (Ka/Ks < 1). Nineteen pairs (43.18%) of ARF genes in fig species had Ka/Ks greater than 1, while 25 pairs (56.82%) had Ka/Ks less than 1 (Table S4), indicating the existence of two evolutionary modes of positive and negative selection in fig, with negative selection being the main evolutionary mode.

Collinearity analysis identified 18 separate orthologous genes between *F. carica* and *F. macrocarpa* as well as between *F. carica* and *F. hispida*, indicating that evolutionary distance between fresh fig and two evergreen *Ficus* species was similar (Fig. 4B). *FcARF20* did not exhibit a collinear relationship with either of the two Ficus species, implying that there might be a unique ARF in Fig evolution.

## Protein interaction network of FcARFs

The ARF protein interacts with other proteins, such as IAA and AUX1 proteins, to regulate downstream gene expressions and exert biological functions. Correlation analysis of ARF proteins and interacting proteins in figs, as well as the gene expression correlation network (*Song et al., 2022*), was analyzed based on gene expressions in the developmental stages of fig fruits (Fig. 5B). A protein interaction network was constructed by STRING, which included 5 ARF proteins and nine other proteins (5 IAA proteins). Expressions of the five FcARFs were positively correlated with each other, co-expression scores were all above 0.7, with FcIAA1, FcBZR1 and FcAUX1 being negatively correlated with expressions of the five FcARFs.

In Fig. 5A, thickness of lines represents protein interaction strengths. IAA8 interacted with five FcARFs, and their expressions were positively correlated, including FcARF15-IAA8. In *Arabidopsis*, interactions of AtARF15 and AtIAA8 proteins respond to early auxin responses and cooperatively regulates lateral root growth, development as well as morphogenesis (*Arase et al., 2012*). There was a strong interaction between ARF6-PIF4, with an expression correlation coefficient of 0.808, implying a positive correlation. Direct inhibition of auxin-responsive gene expressions mediated by interactions of AtPIF4 and AtARF6 in *Arabidopsis* constitutes a new layer in regulatory mechanisms of photoinhibition of auxin-induced hypocotyl elongation (*Oh et al., 2014*). IAA1 was predicted to interact with FcARF8 and FcARF19, and expression correlation indices were −0.582 and −0.571, respectively, implying negative correlations. This is consistent with the

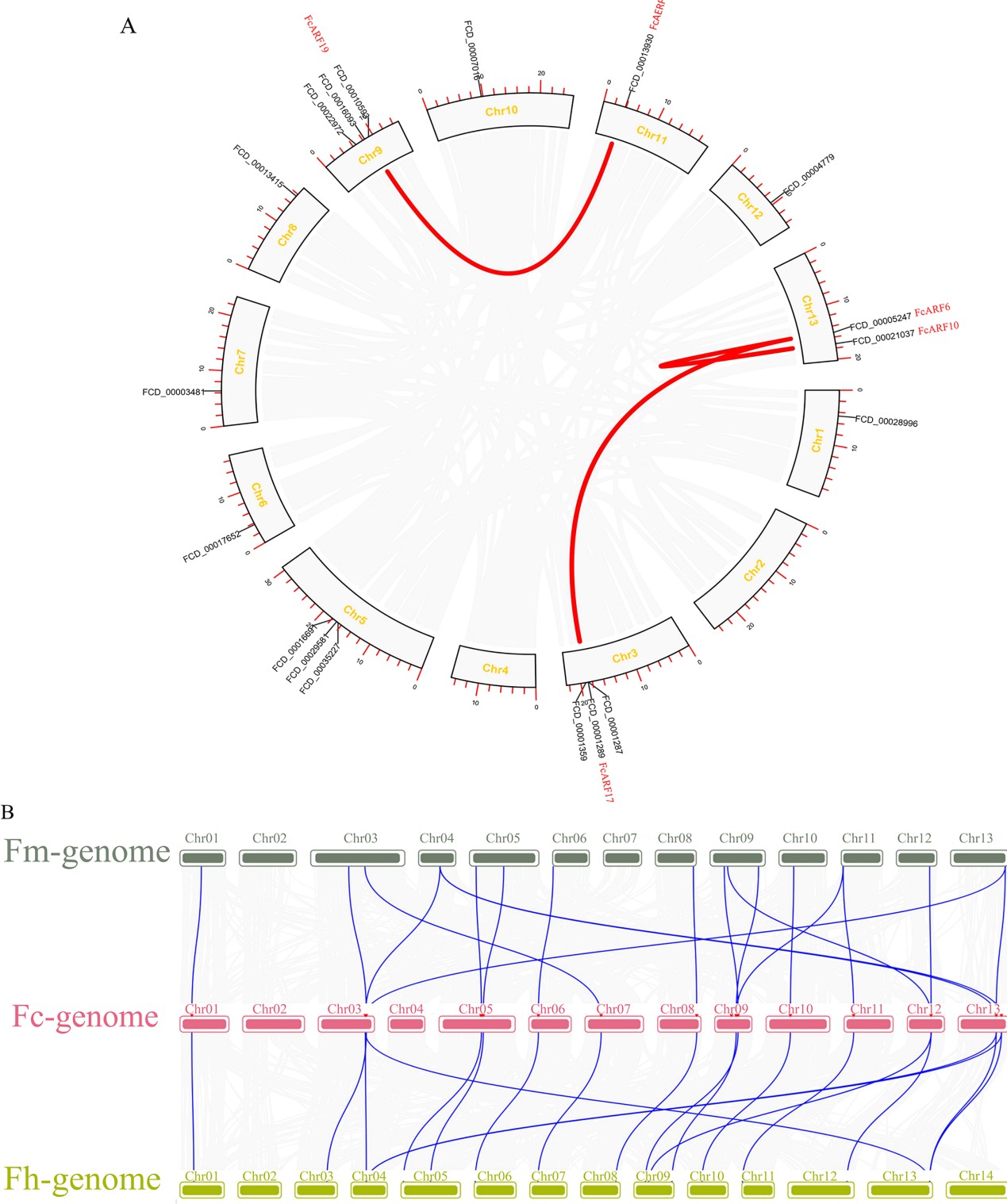

**Figure 4 Chromosomal localization and collinearity analysis of the FcARF gene family.** (A) Collinearity analysis of FcARF genes, the circle plot was created with the MCScanX tool. Identified collared genes are linked by colored lines. (B) Collinearity relationship of ARF genes among *Ficus carica* (Fc), *Ficus hispida* (Fh), and *Ficus microcarpa* (Fm). Identified colinear genes are linked by blue lines.

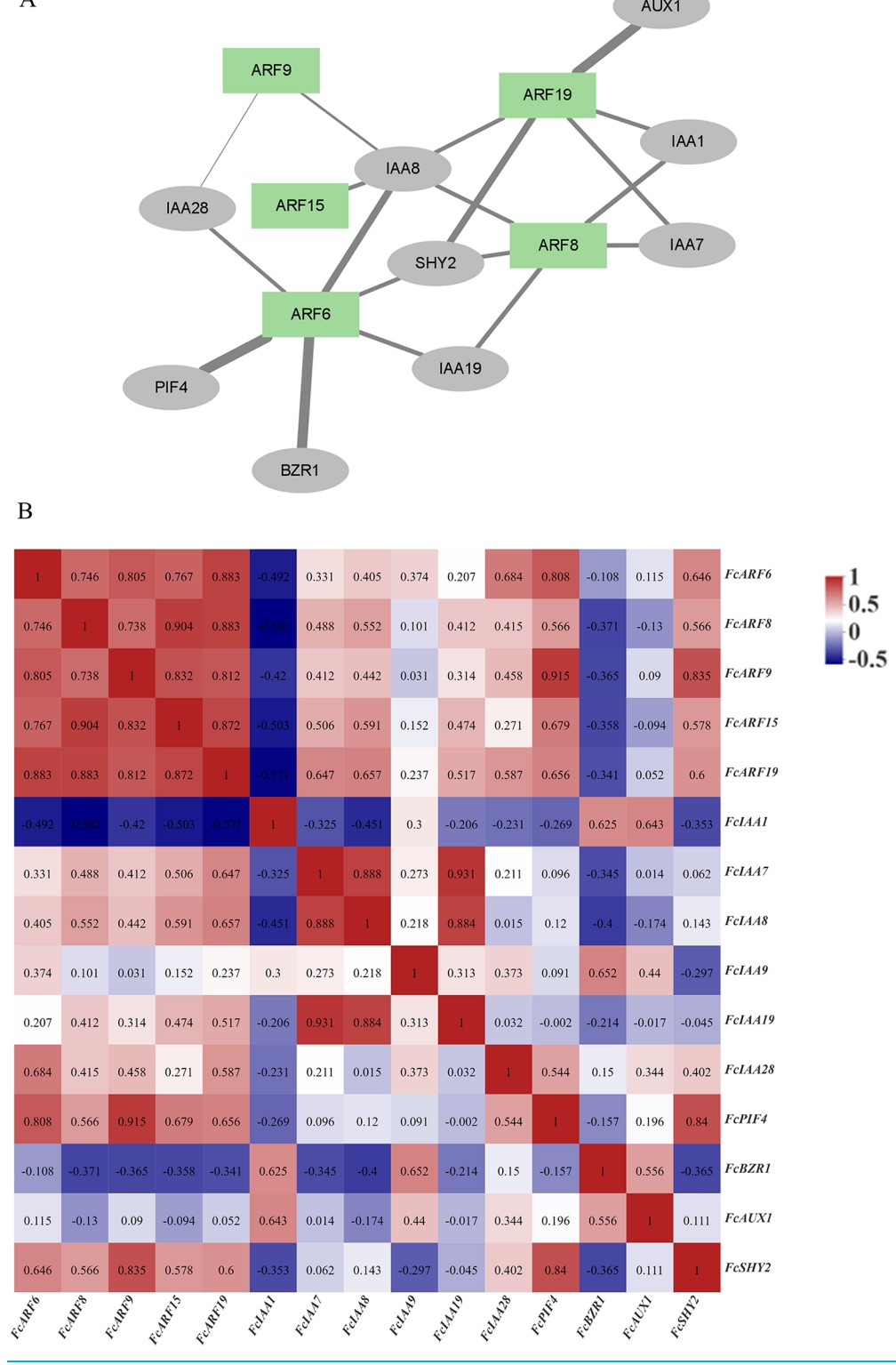

**Figure 5 Correlation analysis network among auxin signaling-related predicted proteins.** (A) The FcARFs and other predicted proteins interaction network were analyzed in the STRING database. Green color indicated FcARFs and the gray color indicated other auxin signaling-related predicted

**Figure 5** (continued)
proteins. A thicker line indicates a stronger interaction between the two proteins. (B) The aux-in signaling-related predicted genes expression correlation analysis was performed by Pearson algorithm, based on these gene's expression of the female flower and peel during the fig fruit development, the data in the cells were indicated the correlations scores.               

prediction that IAA interacts with ARF and negatively regulates its expressions. In the regulatory network composed of fig ARFs, there were strong positive correlations among FcARFs levels, indicating functional complementarity and redundancy among them. The expression levels and interaction coefficients predict that FcIAA8 is a key positive cooperative factor in the network, while IAA1 is a negative regulatory cooperative factor.

## Expressions of *FcARFs* in different tissues

Expression patterns of 20 FcARFs in fig's root (RO), stem (ST), old leaf (OL), young leaf (YL), flower (FL) and peel (PE) were analyzed by qRT-PCR (Fig. 6). Twenty *FcARFs* showed tissue and organ expression specificity. In class-I, *FcARF1/12/14/18/20* were highly expressed in female flowers and peel, the major fruit components. *FcARF 2/3/4/9/11* were highly expressed in the stem. In class-II, *FcARF 5/8/15* were highly expressed in female flowers and peel while *FcARF 6/7/19* were highly expressed in the stem. In class-III, all FcARFs (*FcARF 10/13/16/17*) were highly expressed in female flowers and peel. These findings imply that expressions of these *FcARF* genes are high in reproductive organs, but low in vegetative organs, such as leaves (Fig. 6).

## Expressions of *FcARFs* during fig fruit ripening

Expressions patterns of *FcARFs* in 'Purple peel' fig peel and flower tissues at different growth stages were constructed by TBtools (Fig. 7). Based on FPKM values, they were divided into four groups, among which four members greater than 100 came from three classes, respectively. Class-I had four members (*FcARF2*, *FcARF3*, *FcARF11*, and *FcARF12*) expressed in early stages of flower and peel development. The 50–100 group consists of three members (*FcARF1, FcARF8* and FcARF18) from class-Ia and class-II. Expression patterns of *FcARF1* and *FcARF18* were consistent, being highest in early stages of female flower development. The rest of the members with FPKM values between 10 and 50.

 During the six growth stages of female flowers and peel, *FcARF3* from class-Ic was stably expressed, with FPKM values >50. In addition, expressions of *FcARF3* in fast-growing phase I and III were higher than those in slow-growing phase II. Expressions of class-Ia members *FcARF2/9/11* were highest in the early developmental stages of fruits and were decreased after maturity. *FcARF11* levels were highest at P1, P2 and P4, while *FcARF9* levels were highest at P3. In this study, *FcARF* gene expressions in peel and flower tissues of fig fruit at 6 different stages of ripening levels were evaluated by qRT-PCR (Figs. S5 and S6). Expression profiles of *FcARF* in female flowers and peel tissues obtained from transcription sequencing data should be verified.

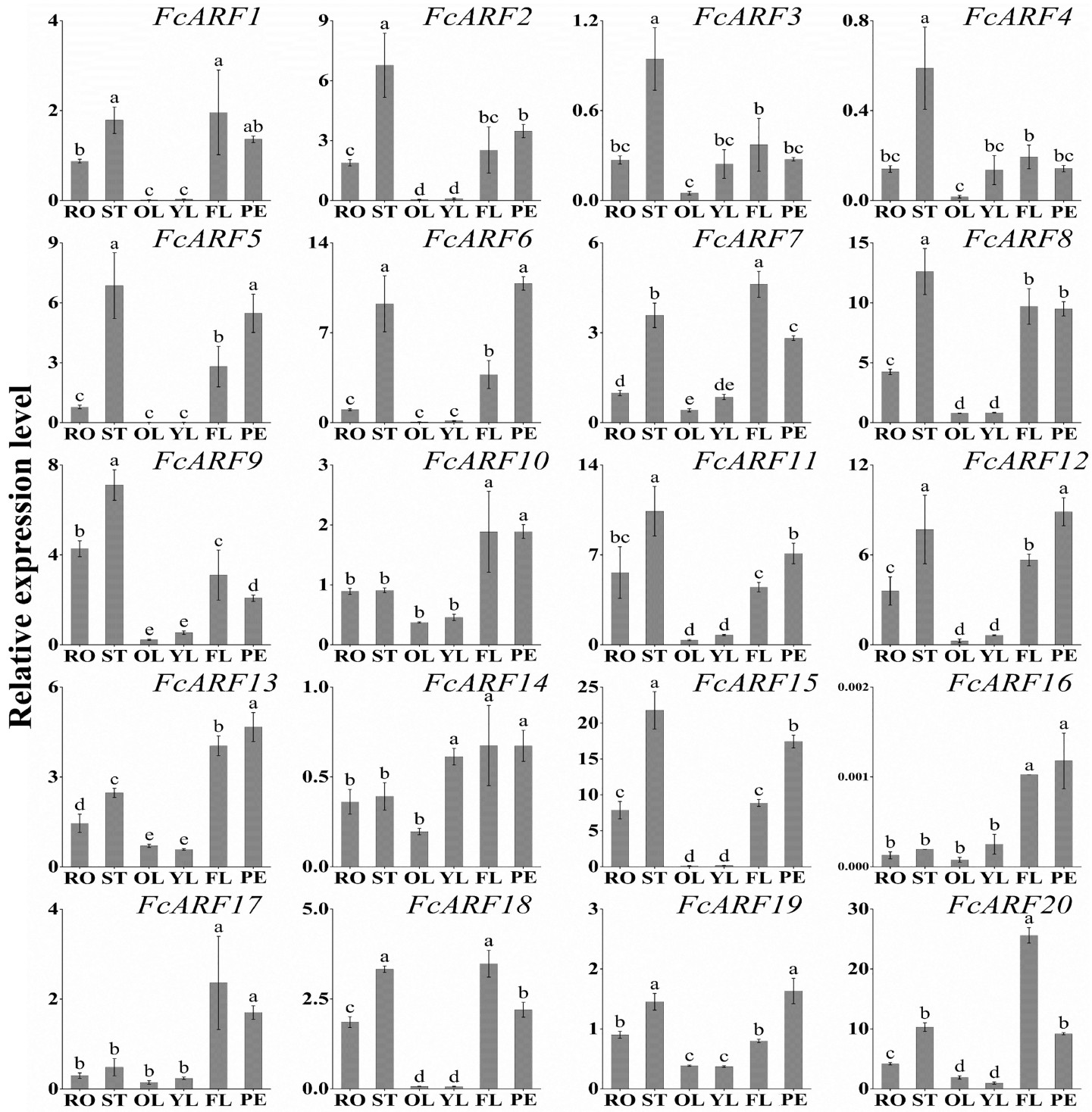

**Figure 6 Tissue-specific transcript expression patterns of 20 ARF genes in the 'Purple Peel' fig.** Different plant tissues. RO,: Root; ST, Stem; OL, Old Leaf; YF, Young Leaf; FL, Flower; PE, Peel. Relative transcript levels are calculated by qRT-PCR with *β-actin* as a standard. Data are means ± SD of three separate measurements. The qRT-PCR data were analyzed by relative quantification using $2^{-\Delta\Delta Ct}$. Different letters above the bars indicate statistically significant differences ($P = 0.05$).

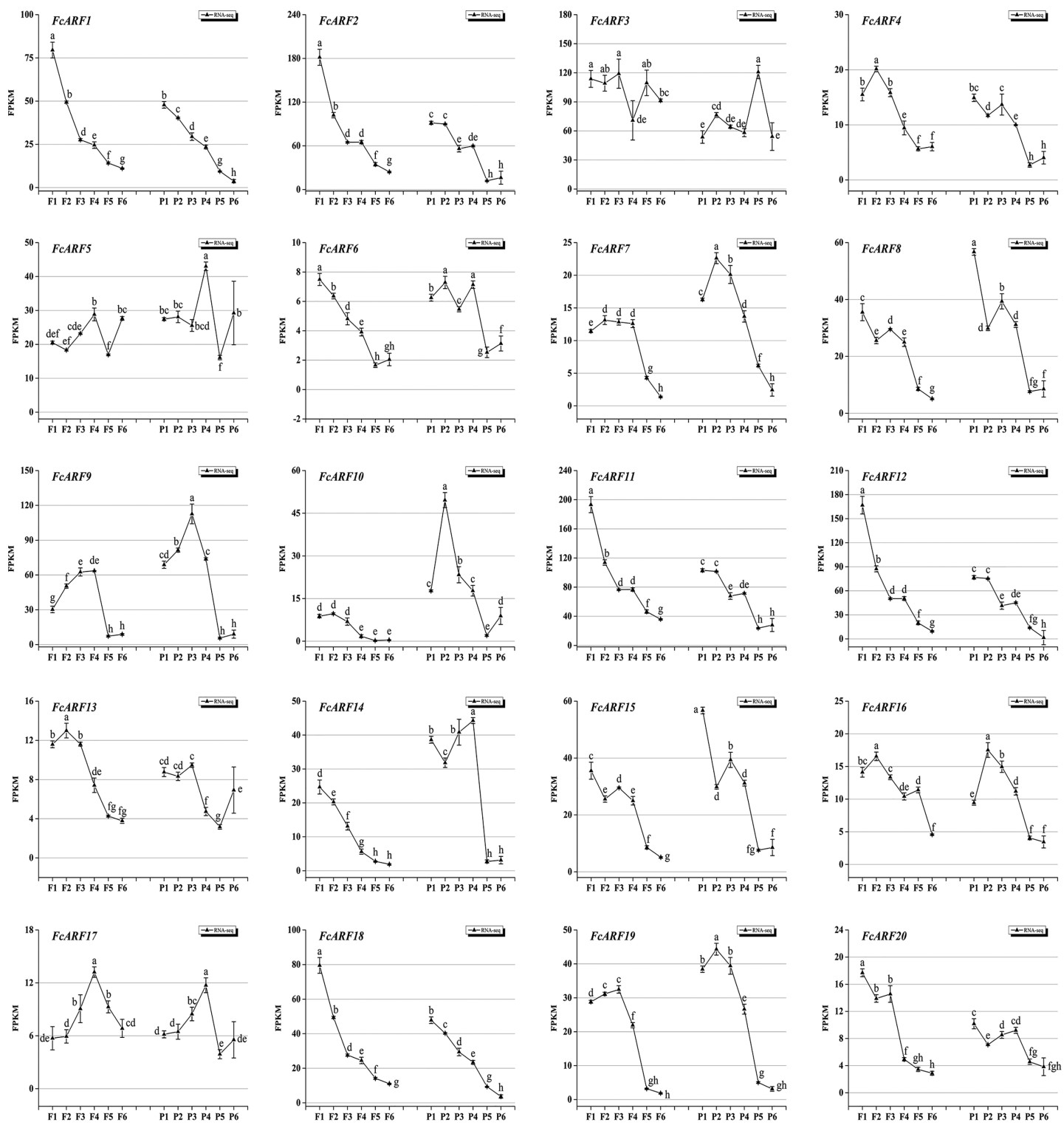

**Figure 7 Transcript expression pattern of 20 ARF genes in the 'Purple Peel' fig during different fruit development stages.** Fruit developmental stages. Stage 1: represented phase I (the first rapid growth period); stages 2, 3, and 4: were the early, middle, and late stages of phase II (slow growth period); stages 5 and 6: represented phase III (the second rapid growth period).

## DISCUSSION

To establish the significance of ARF genes in the auxin signal transduction pathway in fig, we evaluated the main structural characteristics of the ARF gene family in fig. Using genome sequencing results, we isolated and identified 20 fig ARF genes.

### Phylogenetic tree and protein structure predictions of ARF protein function

Phylogenetic trees have been used to predict the functions of different genes (*Pandey et al., 2018*). Figure 1 shows that fig is highly homologous to *Arabidopsis thaliana*, a model plant, with six sister pairs. Phylogenetic tree analysis of ARF proteins in *Arabidopsis* and fig can predict the functions of FcARFs by clarifying their phylogenetic relationships.

For instance, *FcARF3* and *FcARF4* of fig belong to class-Ic with *AtARF3* and *AtARF4*, which may be involved in regulation of flower formation, development and patterns. Class-Ia, which consists of *FcARF1*, *FcARF2*, *FcARF9*, and *FcARF11* may have a role in regulating leaf senescence, floral organ shedding and auxin homeostasis. *FcARF10*, *FcARF13*, *FcARF16* and *FcARF17* belong to class-III with *AtARF10*, *AtARF16* and *AtARF17*, which may be involved in regulation of pollen tube wall formation, root cap formation and seed germination.

Apart from *FcARF18*, which does not have the DBD domain, other FcARFs (*FcARF1–17, 19, 20*) have DBD and MR domains that determine their functions based on amino acid compositions (*Sun et al., 2015*; *Wang et al., 2007*; *Okushima et al., 2005*). Through domain comparisons (Fig. S3 and Table S3), it can be found that *FcARF18* lacks motif 2 on the DBD domain, which inhibits its ability to recognize and bind DNA by forming a helix-turn-helix domain (*Tiwari, Hagen & Guilfoyle, 2003*; *Roosjen, Paque & Weijers, 2018*). In addition, most plant ARF gene families have an ARF protein without the CTD domain. All class-III FcARFs have truncated CTD domains, which is similar in *Arabidopsis thaliana* and other species (Fig. 2A). Responses and signal transduction of plant cells to auxin are mainly mediated by the TIR1/AFB/-Aux/IAA-ARF signaling pathway in the nucleus. ARF participates in the auxin signaling pathway by forming a heterodimer with CTD regions of the Aux/IAA protein. The C-terminal domains of proteins are homologous. Without a complete CTD domain, FcARF10/13/16/17 may not participate in the auxin signaling pathway or perform its independent functions (*Liu et al., 2011*).

The ARF genes are transcriptional activators or inhibitors. The activation domain (AD) in glutamine (Q) and leucine (L) in the middle region of the ARF protein can act as a transcription activator, while the repression domain (RD) rich in proline (P), serine (S) and threonine (T) can act as a transcription inhibitor (*Tiwari, Hagen & Guilfoyle, 2003*; *Zouine et al., 2014*). In this study, the intermediate region of transcription factors in class-Ia and class-II was found to be rich in glutamine (Q), and it is postulated that class-II members have transcriptional activation functions. *AtARF5*, *AtARF6*, *AtARF7*, *AtARF8* and *AtARF19* in *Arabidopsis thaliana*, which are homologous to fig class-II, are ARF[ClassA] transcription activators. Middle regions of classes-I and -III are rich in proline (P), serine

(S) and threonine (T). Therefore, members of class-Ia and class-II inhibit transcription. Our findings should be verified by studies of regulatory mechanisms (*Roosjen, Paque & Weijers, 2018*).

## Evolution and gene structures of the ARF family

Gene replication, including tandem repeat and large fragment replication, is the most important mechanism of gene recombination and amplification (*Vision, Brown & Tanksley, 2000*). Tandem repeat refers to two adjacent genes on the same chromosome, while large fragment replication events occur on different chromosomes, which is an important way of gene expansion and species evolution (*Liu et al., 2011*; *Xing et al., 2011*). Compared to other plants, there are 20 ARF genes in fig, which are less than those in other plants, such as 23 in *Arabidopsis thaliana*, 25 in rice and 39 in poplar. Combined with collinearity analysis, there are probably no large-scale gene repetition events in fig evolution. Collateral gene pairs (*FcARF13* and *FcARF10*) were located on chromosome 1, while *FcARF19*, *FcARF20*, *FcARF13* and *FcARF17* were located on chromosomes 9, 11, 13 and 3, respectively. The former belongs to tandem replication, while the latter belongs to large fragment replication. In this study, 3 (15%) of 20 FcARFs were established to be involved in tandem repeat or large fragment replication events, while 33% of MdARFs in apples were associated with gene replication (*Luo et al., 2014*). Even though there are few gene replication events in fig, we postulated that gene replication is the main mode of ARF gene family amplification, implying that different types of ARFs gene family members have different evolutionary paths and rates.

FcARFs are mainly involved in plant hormone responses, growth regulation, light response and environmental stress responses. As trans-acting factors, hormones can bind hormone response elements in promoter sequences, thereby regulating the transcriptional activities of target genes, and can participate in plant growth as well as environmental stress processes, such as salt and temperature as signal transduction factors (*Gunes et al., 2007*). We found that the ARF gene family has various hormone-responsive elements, hat hormones induce ARF gene family expressions in fig, participating in plant growth and development.

## FcARFs play important roles in fig fruit development

Expression levels of ARF family members in different parts of fig were analyzed to elucidate the function of *FcARFs*. There were spatial expressions of *FcARFs* in roots, stems, old leaves, young leaves, female flowers and peel at the transcription level. Most ARF genes were highly expressed in female flowers and peels, followed by stems and roots, and least expressed in old leaves, indicating that ARF family members have important roles in fig fruit growth and development. Besides, *AtARF1* is involved in floral organ abscission (*Ellis et al., 2005*), and its homologous gene (*FcARF1*) is highly expressed in female flower tissues, implying a role in flower growth. Moreover, *FcARF*s (*FcARF 1/2/3/4/5/6/7/8/9/11/ 12/14/15/18/19/20*) exhibited tissue-specific transcript accumulation patterns. It has been reported that *AtARF11* is highly expressed in the roots and promotes lateral root formation (Table S1); (*Chen et al., 2021*; *de Jong et al., 2015*; *Diao et al., 2020*; *Ellis et al., 2005*; *Ge et al.,*

*2016*; *Goetz et al., 2006*; *Kelley et al., 2012*; *Krogan et al., 2014*; *Kumar, Stecher & Tamura, 2016*; *Li et al., 2016*; *Lim et al., 2010*; *Liu et al., 2015*; *Luo, Zhou & Zhang, 2018*; *Okushima et al., 2005*; *Pekker, Alvarez & Eshed, 2005*; *Sagar et al., 2013*; *Sessions et al., 1997*; *Shen et al., 2010*; *Song et al., 2015*; *Wan et al., 2014*; *Wang et al., 2021*, *2005*, *2010*, *2018*; *Yang et al., 2013*; *Ye et al., 2016*; *Zhang et al., 2020*, *2021*, *2015*). *FcARF11* has a high homology with *AtARF11* and has high transcript accumulations in stem and roots (Fig. 6), indicating that the roles of ARF during root development in fig are similar to those in *Arabidopsis thaliana*. ARF7/ARF19 mutants in *Arabidopsis* exhibit strong auxin-related phenotypes, including during severe damage to root formation (*Okushima et al., 2007*; *Okushima et al., 2005*). AtARF19 are homologous and have high transcript expressions in roots. It has been postulated that *FcARF19* has important roles in root formation and fig development.

Fruit development and ripening is a coordinated process of reproductive organ cell division, differentiation and expansion that is regulated by transcriptional regulatory networks (*Uchiumi & Okamoto, 2010*). Auxin promotes the growth of female flowers as well as receptacles and causes fruit enlargement that plays a key role in fruit ripening (*Flaishman, Rodov & Stover, 2008*; *Kumar, Tyagi & Sharma, 2011*). In this study, all *FcARFs* were expressed from the first stage (phase I) to sixth stage (end of phase III) of fruit development. The fruits of figs went through a fast-slow-fast developmental period. Sensing and transmitting signals have been shown to reveal changes in auxin expression patterns (*Estrada-Johnson et al., 2017*). *AtARF2* has an important role in regulating flower formation as well as senescence of *Arabidopsis thaliana* (*Ellis et al., 2005*). Its homologous gene (*FcARF2*) was highly expressed in the early flower and peel development stages, which may regulate fig flower morphologies. *FcARF11* and *FcARF12* from class-Ia were abundantly expressed during the early stages of fig fruit female flower as well as peel development and were predicted to function as transcriptional repressors (Fig. 2B). It has not been established whether FcARF inhibition acts synergistically with other TFs.

*AtARF3* has a major role in nutrition and reproductive development, regulation of gene and embryonic development, and determining the morphological patterns of pistil (*Harper et al., 2000*; *Sessions et al., 1997*; *Xiao et al., 2004*). Its homologous gene (*FcARF3*) is highly expressed during whole growth periods of fig fruits, and suppressed in middle slow growth periods, in tandem with the fig fruit development curve (*Song et al., 2021*). Moreover, it may be involved in various reproductive and developmental processes, such as female flower development in figs. The fig fruit development candidates obtained from fig ARF gene family analysis is the basis for the next step in fruit development, such as expansion mechanisms caused by female flower development.

## CONCLUSIONS

The ARF genes are mainly expressed in fruits and flowers and are involved in development and morphogenesis. The common fig fruit contains enlarged female flowers that are closely correlated with development and maturity. Twenty FcARFs containing similar conserved motif structures were identified from fig genome databases, and were divided into three classes (I, II and III) by phylogenetic analyses. They were differentially expressed

in tissues, with *FcARF1/5/8/10/12/13/1415/16/17/18/20* being highly expressed in female flowers and peels, indicating their possible importance in fruit development. Further analysis of the *FcARFs* expression levers in female flower and peel during common fig *cv*. Purple Peel fruit development. *FcARF2*, *FcARF11* and *FcARF12* are class-Ia members that are mainly involved in regulating flower development. They were highly expressed in early stages of female flower development. Expression patterns of *FcARF3* were in line with fig fruit development curve, implying that it may be involved in the rate-limiting process of fig fruit development as a repressor. This study forms the basis for further functional verification of the fig ARF gene family.

### Funding
This study was supported by the National Natural Science Foundation of China (31372007). The funders had no role in study design, data collection and analysis, decision to publish, or preparation of the manuscript.

### Grant Disclosures
The following grant information was disclosed by the authors:
National Natural Science Foundation of China: 31372007.

### Competing Interests
The authors declare that they have no competing interests.

### Author Contributions
- Haomiao Wang conceived and designed the experiments, performed the experiments, analyzed the data, prepared figures and/or tables, and approved the final draft.
- Hantang Huang conceived and designed the experiments, analyzed the data, prepared figures and/or tables, and approved the final draft.
- Yongkai Shang performed the experiments, prepared figures and/or tables, and approved the final draft.
- Miaoyu Song conceived and designed the experiments, analyzed the data, prepared figures and/or tables, authored or reviewed drafts of the article, and approved the final draft.
- Huiqin Ma conceived and designed the experiments, authored or reviewed drafts of the article, and approved the final draft.

### Data Availability
The raw data is available in the Supplemental File.

### Supplemental Information
Supplemental information for this article can be found online at http://dx.doi.org/10.7717/peerj.13798#supplemental-information.

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
