# Peer review of "Identification and characterization of auxin response factor (ARF) family members involved in fig (Ficus carica L.) fruit development"

_PeerJ, doi:10.7717/peerj.13798_

## Round 0.1 · original submission · Major Revisions

Authors should consider reviewers' feedback, specially R2 and R3.

Reviewer 1 ·

Basic reporting

The MS by Haomiao Wang focuses on identification of ARF gene family in fig followed by the study of phylogeny, gene structures analysis, conserved motif analysis etc. This study provides comprehensive information for the fig ARF gene family.

Experimental design

These are superficial in silico analyses, although the author also analyzed the expression analysis of these genes by QRT-PCR. If necessary, functional verification through genetic modification is required.

Validity of the findings

More comments were listed below:
1. The Latin names of some species are not italicized. Eg. L128, L141.
2. The writing of this manuscript needs improvement: tiny errors, eg. L320 (missing space); L146 (extra space). Language needs to be promoted by fluent English speakers.
3. The legends of figures and tables are not written in detail. Ensure that the reader can get all the information (eg. Softwares, parameters, abbreviation…) displayed by the figure without the main text.
4. Some software require references to be cited (eg. L171, STRING; L146, gsds; L123, PFAM; L127, SMART).

·

Basic reporting

1. There are so many ambiguous word in this manuscript, such as “some FcARFs”(Line 289).
2. In the Table 2, authors wrote “Table 1” (Page 35).
3. The written English should be improved.

Experimental design

1. The abbreviation and full name of tissues must be clearly stated. (Line 288).
2. In the “Expression analysis of FcARFs“part, authors failed to state the accurate methods. We can’t know which data was acquired and calculated by qRT-PCR. And which part was analyzed using database data. (Line 176).

Validity of the findings

1. The molecular weight differ greatly in the Table 2 (from 52.72 kDa to 127.056kDa), why?
2. The FcARF4 (768 aa) coding only 52.73 kDa protein, why the smaller protein, such as FcARF10 (588 aa), coding bigger protein (65.32 kDa)? The information of this table must be verified once again.
3. In the Figure 1, I suggest that “Ia, Ib, Ic, III, III” should be replaced by “class-Ia, class-Ib, class-Ic, class-III, class-III”.
4. I can’t get any gene structures in the Figure 2(A). This part only give us a phylogenetic tree information, and this tree should be done by amino acid sequences. Furthermore, the data in Figure 2(A) and Figure 2(B) is repeated.
5. The phylogenetic tree has been shown in the Figure 2, it should not be stated in the Figure 3.
6. I suggest the “conserved motif” should be replaced by “Introns and exons” or some other words. Because we can’t find the conserved motif easily in this picture, I think.
7. Authors should label the number of base pair in the figure Figure 2(B), such as -2000, -1000 ------0.
8. Why you analyzed the relationship of bHLH in the Figure 4(B)?
9. The ARF should be revised to FcARF in the Figure 5. Moreover, authors should be state which protein interact with each other first. Then authors could tell us how these interacted proteins regulated. And the Figure 5 should be redesigned.
10. The “Relative expression level” should be labeled in the Figure 6.
11. Authors show us the expression data by the Classes (Line300), but the data was shown by the gene family number. I suggest authors replacerep the data by Classes in the Figure 6.

Additional comments

No additional comments.

Reviewer 3 ·

Basic reporting

The English language should be improved to ensure that an international audience can clearly understand your text. Some examples where the language could be improved include lines 16-32-33- -112-151-152-153 -164-165-166-167-168 -172- 317.
The current phrasing makes comprehension difficult. I suggest you have a colleague who is proficient in English and familiar with the subject matter review your manuscript or contact a professional editing service.

Experimental design

No comments

Validity of the findings

No comments

Additional comments

Lines 167-168: This sentence needs improvement.
Line 173: Which database?
Line 177-178: This part is not clear.
Line 284: What is IAA1 ? You mentioned it for the first time so it should be spelled out.
Line 284: What are AUX1 and IAA8? You mentioned it for the first time so it should be spelled out
Line 290: These results indicated that these five genes: Which genes?
Line 313: What’s: It is not academic.
Line 341: can’t: Not academic
Line 349: the meaning not clear.
Line 359: so: Not academic.
380: the meaning not clear.
398: The structure of the sentence is not right.
Line 414: AtARF2 plays
Figure 2 B: first line: are FcARF 10, 13, 16 contain CTD or not? Revise.
Figure 4: in the caption you mentioned (bHLH) why? Revise.
Figure 5: which boxes represent IAA transcription factors, green or red? Revise.
Protein interaction network of FcARFs
In this part, there is a new information to the reader that needs more clarification. For example: what is IAA1? why you mentioned it? Does it have role in Auxin response or signaling? Provide more details here, please.

---

## Round 0.2 · Minor Revisions

Authors should check minor observations.

Reviewer 3 ·

Basic reporting

The manuscript has improved greatly after revisions.

Experimental design

no comment

Validity of the findings

no comment

Additional comments

line 108: (has been) was written twice, please revise
line 191: genepairs change to gene pairs

---

## Round 0.3 · Minor Revisions

Before a final decision, authors must address 2 issues:

1- The FcARFs tree, whose method is explained in item 2.3, should be better detailed. Authors must detail which phylogenetic method was used for the tree, how gaps were treated and how many positions were used for the tree. In MEGA, this information is usually provided in "caption".

2- Authors must display a tree with bootstraps and distance scale as a supplementary material.

3- Table 1 should be moved to the supplementary material.

These changes are crucial before a final decision.

Reviewer 3 ·

Basic reporting

no comment

Experimental design

no comment

Validity of the findings

no comment

Additional comments

no comment

---

## Round 0.4 · accepted · Accept

The authors have addressed all concerns raised by editors and reviewers.